# Monoterpenoid Glycosides from the Leaves of *Ligustrum robustum* and Their Bioactivities

**DOI:** 10.3390/molecules27123709

**Published:** 2022-06-09

**Authors:** Shi-Hui Lu, Jing Huang, Hao-Jiang Zuo, Zhong-Bo Zhou, Cai-Yan Yang, Zu-Liang Huang

**Affiliations:** 1College of Pharmacy, Youjiang Medical University for Nationalities, Baise 533000, China; zzb7855@163.com (Z.-B.Z.); yjsyangcaiyan@163.com (C.-Y.Y.); 2Key Laboratory of Drug Targeting, Ministry of Education, West China School of Pharmacy, Sichuan University, Chengdu 610041, China; 3Department of Laboratory Science of Public Health, West China School of Public Health, Sichuan University, Chengdu 610041, China; zuohaojiang@scu.edu.cn

**Keywords:** *Ligustrum robustum*, monoterpenoid glycoside, FAS, *α*-glucosidase, antioxidant, anti-obesity, hypoglycemic

## Abstract

The leaves of *Ligustrum robustum* have been applied as Ku-Ding-Cha, a functional tea to clear heat, remove toxins, and treat obesity and diabetes, in Southwest China. The phytochemical research on the leaves of *L. robustum* led to the isolation and identification of eight new monoterpenoid glycosides (**1**–**8**) and three known monoterpenoid glycosides (**9**–**11**). Compounds **1**–**11** were tested for the inhibitory activities on fatty acid synthase (FAS), *α*-glucosidase, *α*-amylase, and the antioxidant effects. Compound **2** showed stronger FAS inhibitory activity (IC_50_: 2.36 ± 0.10 μM) than the positive control orlistat (IC_50_: 4.46 ± 0.13 μM), while compounds **1**, **2**, **5** and **11** displayed more potent ABTS radical scavenging activity (IC_50_: 6.91 ± 0.10~9.41 ± 0.22 μM) than the positive control L-(+)-ascorbic acid (IC_50_: 10.06 ± 0.19 μM). This study provided a theoretical basis for the leaves of *L. robustum* as a functional tea to treat obesity.

## 1. Introduction

Ku-Ding-Cha has been used widely as a functional tea to clear heat, remove toxins, and treat obesity, diabetes and so on, in Southwest China for a long time [1,2]. It was produced from the leaves of more than 30 plants from 13 genera in 12 families, in which the most common categories were from the genus *Ligustrum* (Oleaceae) and the genus *Ilex* (Aquifoliaceae) [3]. *Ligustrum robustum* (Roxb.) Blume, distributed widely in Southwest China, India, Burma, Vietnam and Cambodia, has been consumed as Ku-Ding-Cha in Southwest China, especially in Guizhou Province [4]. *L. robustum* has been classified as a food by the Chinese Ministry of Health since 2011 [5]. In the past two decades, the phytochemical studies on *L. robustum* led to the isolation and identification of monoterpenoid glycosides, phenylethanoid glycosides, iridoid glycosides, flavonoid glycosides and triterpenoids [1,6,7,8,9,10,11]. The biological research on *L. robustum* reported the anti-obesity activity of the total glycosides and the aqueous extract [2,5], the antioxidative, anti-inflammatory and hepato-protective effects of the aqueous extract [4], and the antioxidant effect of some constituents [1,10]. In our previous study on *L. robustum* [12], some antioxidative and *α*-glucosidase inhibitory components, which might be a part of anti-diabetic ingredients of *L. robustum* [13,14,15,16], were discovered. However, to the best of our knowledge, the exact anti-obesity ingredients of *L. robustum* and their mechanisms are still unclear so far.

Studies revealed that fatty acid synthase (FAS) catalyzed the synthesis of saturated long-chain fatty acids from acetyl-coenzyme A, malonyl-CoA and NADPH; FAS expressed high in normal adipose, liver tissues, lactating mammary glands, and in patient tumor tissues at later stages of disease, while most normal tissues showed low levels of FAS expression [17,18,19]. Thus, FAS is a potential therapeutic target for anti-obesity and anti-cancer drugs. There have been no reports on the screening FAS inhibitors from the constituents of *L. robustum*. In this work, eight new monoterpenoid glycosides, named ligurobustosides T (**1**), T_1_ (**2**), T_2_ (**3**), T_3-4_ (**4**), T_5_ (**5**), T_6_ (**6**), T_7_ (**7**), T_8–9_ (**8**), and three known monoterpenoid glycosides (**9**–**11**) (Figure 1) were isolated from the leaves of *L. robustum*. This paper deals with the isolation and structure elucidation of **1**–**11**, and it describes their inhibitory activities on FAS, *α*-glucosidase, *α*-amylase, and their antioxidant effects.

## 2. Material and Methods

### 2.1. General Experimental Procedure

First, 1D and 2D NMR spectra were measured on a Bruker Ascend^TM^ 400 NMR spectrometer (Bruker, Germany) (^1^H at 400 MHz, ^13^C at 100 MHz) or an Agilent 600/54 Premium Compact NMR spectrometer (Agilent, Santa Clara, CA, USA) (^1^H at 600 MHz, ^13^C at 150 MHz) with CD_3_OD as the solvent at 25 °C. Chemical shifts are expressed in *δ* (ppm) with tetramethylsilane (TMS) as the internal standard, and coupling constants (*J*) are reported in Hz. High-resolution electrospray ionization mass spectroscopy (HRESIMS) was carried out on a Waters Q-TOF Premier mass spectrometer (Waters, Milford, MA, USA). The IR absorption spectrum was measured with a PerkinElmer Spectrum Two FT-IR spectrometer (PerkinElmer, Waltham, MA, USA). UV spectrum was recorded using a UV2700 spectrophotometer (Shimadzu, Kyoto, Japan). Optical rotation value was analyzed with an AUTOPOL VI automatic polarimeter (Rudolph, Hackettstown, NJ, USA).

UV-vis absorbance was analyzed with a Spark 10M microplate reader (Tecan Trading Co. Ltd., Shanghai, China). Preparative HPLC was performed on a GL3000-300 mL system instrument (Chengdu Gelai Precision Instruments Co., Ltd., Chengdu, China) with a GL C-18 column (particle size 5 μm, 50 × 450 mm) and a UV-3292 detector operating at 215 nm, eluting with MeOH-H_2_O at a flow rate of 30 mL/min. Column chromatography (CC) was performed on silica gel (SiO_2_: 200–300 mesh, Qingdao Ocean Chemical Industry Co., Qingdao, China), MCI-gel CHP-20P (75–150 μm, Mitsubishi Chemical Co., Tokyo, Japan), and polyamide (60–90 mesh, Jiangsu Changfeng Chemical Industry Co., Yangzhou, China). TLC was carried out on precoated HPTLC Fertigplatten Kieselgel 60 F_254_ plates (Merck), and the spots were visualized by spraying with *α*-naphthol-sulfuric acid solution or 10% sulfuric acid ethanolic solution and heating at 105 °C for 2–5 min. NADPH and acetyl-coenzyme A (Ac-CoA) were obtained from Zeye Biochemical Co., Ltd. (Shanghai, China). Methylmalonyl coenzyme A tetralithium salt hydrate (Mal-CoA) was purchased from Sigma-Aldrich (St. Louis, MO, USA). 2,2-Diphenyl-1-picrylhydrazyl (DPPH) was obtained from Macklin Biochemical Co., Ltd. (Shanghai, China). 2,2′-Azino-bis(3-ethylbenzthiazoline-6- sulphonic acid) ammonium salt (ABTS) was purchased from Aladdin Industrial Co., Ltd. (Shanghai, China).

### 2.2. Plant Material

The leaves of *L. robustum* were collected from Yibin City, Sichuan Province, China, in April 2017, and identified by Professor Guo-Min Liu (Kudingcha Research Institute, Hainan University, Haikou, 570228, China). A voucher specimen (No. 201704lsh) was deposited in West China School of Pharmacy, Sichuan University, China.

### 2.3. Extraction and Isolation

The fresh leaves of *L. robustum* were stirred and dried at 120 °C for 50 min and then powdered. The dried raw powder (7.0 kg) was extracted under reflux with 70% ethanol (28 L × 1) in a multi-function extractor for 2 h. The ethanol extract was filtrated and concentrated in vacuo to obtain a dark brown paste (2.2 kg). The paste was dissolved in 95% ethanol (3 L), and then, the distilled water (3 L) was added to precipitate the chlorophyll. After filtration, the filtrate was concentrated in vacuo to gain a brown residue (1.0 kg). The residue was chromatographed on silica gel column, eluting with CH_2_Cl_2_-MeOH (10:0–0:10), to yield Fr. I (84 g), Fr. II (145 g), Fr. III (93 g), and Fr. IV (70 g). Fr. II was separated repeatedly by CC on silica gel, eluting with CH_2_Cl_2_-MeOH-H_2_O (200:10:1–40:10:1) or EtOAc-MeOH-H_2_O (50:2:1–50:3:1), and then subjected to polyamide column (EtOH-H_2_O, 0:10–7:3) and MCI column (MeOH-H_2_O, 3:7–7:3), and purified finally by preparative HPLC (MeOH-H_2_O, 40:60–65:35) or silica gel column (EtOAc- MeOH-H_2_O, 50:2:1–50:3:1), to yield **1** (48.5 mg), **2** (49.2 mg), **3** (11.8 mg), **4** (15.3 mg), **5** (8.2 mg), **6** (17.6 mg), **7** (8.2 mg), **8** (10.7 mg), **9** (20.0 mg), **10** (135.8 mg) and **11** (38.6 mg).

Compound **1**: yellowish amorphous powder. [*α*]^20^_D_-91.9 (*c* 0.27, MeOH); UV (MeOH) λ_max_: (log ε) 214 (4.2), 244 (4.1), 331 (4.4) nm; IR (film) ν_max_: 3375, 2923, 1694, 1630, 1601, 1515, 1446, 1376, 1261, 1025, 928, 836, 811 cm^−1^; ^1^H NMR (CD_3_OD, 400 MHz) data, see Table 1; ^13^C NMR (CD_3_OD, 100 MHz) data, see Table 2; HRESIMS *m*/*z* 647.2679 [M + Na]^+^ (calculated for C_31_H_44_NaO_13_, 647.2680).

Compound **2**: white amorphous powder. [*α*]^23^_D_-29.9 (*c* 0.98, MeOH); UV (MeOH) λ_max_ (log ε): 209 (3.9), 230 (3.9), 314 (4.4) nm; IR (film) ν_max_: 3375, 2927, 1689, 1632, 1604, 1515, 1445, 1263, 1169, 1037, 832 cm^−^^1^; ^1^H NMR (CD_3_OD, 400 MHz) data, see Table 1; ^13^C NMR (CD_3_OD, 100 MHz) data, see Table 2; HRESIMS *m*/*z* 631.2728 [M + Na]^+^ (calculated for C_31_H_44_NaO_12_, 631.2730).

Compound **3**: white amorphous powder. [*α*]^25^_D_-11.0 (*c* 0.47, MeOH); UV (MeOH) λ_max_ (log ε): 208 (3.9), 230 (3.9), 316 (4.4) nm; IR (film) ν_max_: 3370, 2926, 2855, 1696, 1605, 1514, 1448, 1262, 1169, 1036, 833 cm^−^^1^; ^1^H NMR (CD_3_OD, 600 MHz) data, see Table 1; ^13^C NMR (CD_3_OD, 100 MHz) data, see Table 2; HRESIMS *m*/*z* 647.2680 [M + Na]^+^ (calculated for C_31_H_44_NaO_13_, 647.2680).

Compound **4**: white amorphous powder. [*α*]^23^_D_-20.9 (*c* 0.31, MeOH); UV (MeOH) λ_max_ (log ε): 208 (3.9), 229 (3.9), 315 (4.4) nm; IR (film) ν_max_: 3369, 2924, 2854, 1695, 1632, 1604, 1515, 1448, 1262, 1170, 833 cm^−^^1^; ^1^H NMR (CD_3_OD, 400 MHz) data, see Table 1; ^13^C NMR (CD_3_OD, 100 MHz) data, see Table 2; HRESIMS *m*/*z* 665.2784 [M + Na]^+^ (calculated for C_31_H_46_NaO_14_, 665.2785).

Compound **5**: white amorphous powder. [*α*]^23^_D_-29.3 (*c* 0.16, MeOH); UV (MeOH) λ_max_ (log ε): 209 (3.9), 228 (3.9), 315 (4.4) nm; IR (film) ν_max_: 3375, 2926, 1694, 1632, 1605, 1515, 1377, 1262, 1169, 1038, 833 cm^−^^1^; ^1^H NMR (CD_3_OD, 400 MHz) data, see Table 1; ^13^C NMR (CD_3_OD, 100 MHz) data, see Table 2; HRESIMS *m*/*z* 661.2833 [M + Na]^+^ (calculated for C_32_H_46_NaO_13_, 661.2836).

Compound **6**: white amorphous powder. [*α*]^23^_D_-71.0 (*c* 0.35, MeOH); UV (MeOH) λ_max_ (log ε): 209 (3.9), 230 (3.9), 313 (4.4) nm; IR (film) ν_max_: 3410, 2973, 1696, 1604, 1515, 1381, 1260, 1168, 1046, 834 cm^−^^1^; ^1^H NMR (CD_3_OD, 400 MHz) data, see Table 1; ^13^C NMR (CD_3_OD, 150 MHz) data, see Table 2; HRESIMS *m*/*z* 777.3312 [M + Na]^+^ (calculated for C_37_H_54_NaO_16_, 777.3310).

Compound **7**: white amorphous powder. [*α*]^23^_D_-71.0 (*c* 0.35, MeOH); UV (MeOH) λ_max_ (log ε): 208 (3.9), 230 (3.9), 316 (4.4) nm; IR (film) ν_max_: 3410, 2973, 1696, 1604, 1515, 1381, 1260, 1168, 1046, 834 cm^−^^1^; ^1^H NMR (CD_3_OD, 400 MHz) data, see Table 1; ^13^C NMR (CD_3_OD, 150 MHz) data, see Table 2; HRESIMS *m*/*z* 777.3312 [M + Na]^+^ (calculated for C_37_H_54_NaO_16_, 777.3310).

Compound **8**: white amorphous powder. [*α*]^23^_D_-17.8 (*c* 0.21, MeOH); UV (MeOH) λ_max_ (log ε): 208 (3.9), 230 (3.9), 316 (4.4) nm; IR (film) ν_max_: 3391, 2925, 1697, 1632, 1605, 1515, 1446, 1264, 1169, 1041, 834 cm^−^^1^; ^1^H NMR (CD_3_OD, 400 MHz) data, see Table 1; ^13^C NMR (CD_3_OD, 100 MHz) data, see Table 2; HRESIMS *m*/*z* 661.2831 [M + Na]^+^ (calculated for C_32_H_46_NaO_13_, 661.2836).

### 2.4. Acid Hydrolysis of Compounds ***1**–**8***

Compounds **1**–**8** (2 mg) in MeOH (0.1 mL) were added to 1 M H_2_SO_4_ aqueous solution (2 mL) and heated in 95 °C water bath for 6 h, respectively. The hydrolyzed solution was neutralized with 1 M Ba(OH)_2_, filtered and concentrated to a small amount. The monosaccharides in the concentrated solution were identified by TLC with authentic samples, developing with EtOAc-MeOH-HOAc-H_2_O (8:1:1:0.7, 2 developments). The *R_f_* values of d-glucose, d-mannose and l-rhamnose were 0.43, 0.46 and 0.73, respectively.

### 2.5. Enzymatic Hydrolysis of Compounds ***1**–**2***

Compound **1** or **2** (20 mg) was hydrolyzed with cellulase (30 mg) in HOAc-NaOAc buffer solution (pH 5.0, 12 mL) at 37 °C for 12 h. The hydrolyzed product was extracted with Et_2_O and purified on silica gel column (eluting with CH_2_Cl_2_), to give (*R*)-linalool and (*S*)-linalool (4:6) confirmed by [*α*]^27^_D_ +3.5 (*c* 0.09, EtOAc) or +2.8 (*c* 0.07, EtOAc).

### 2.6. Determination of Bioactivities

The inhibitory activities on FAS, *α*-glucosidase and *α*-amylase, and the DPPH and ABTS radical scavenging effects of compounds **1**–**11** were evaluated according to the methods described in the literature [12,18,20], while orlistat, acarbose and L-(+)-ascorbic acid were used as the positive controls, respectively (S1).

### 2.7. Statistical Analyses

Statistical analyses were carried out on GraphPad Prism 5.01. All samples were measured in triplicate. The IC_50_ (the final concentration of sample needed to inhibit 50% of enzyme activity or scavenge 50% of free radical) was obtained by plotting the inhibition or scavenging percentage of each sample against its concentration. The results are reported as mean ± standard deviation (SD). Differences of means between groups were analyzed by one-way analysis of variance (ANOVA) on statistical package SPSS 13.0. The differences between groups were believed to be significant when *p* < 0.05.

## 3. Results and Discussion

### 3.1. Identification of Compounds ***1**–**11***

Compound **1** was analyzed as C_31_H_44_O_13_ by HRESIMS (*m*/*z* 647.2679 [M + Na]^+^, calculated 647.2680 for C_31_H_44_NaO_13_). The ^1^H NMR spectrum of **1** (Table 1) revealed the following signals: (1) a 3,4-disubstituted phenyl at *δ*_H_ 7.05 (1H, d, *J* = 2.0 Hz), 6.95 (1H, dd, *J* = 8.0, 2.0 Hz) and 6.77 (1H, d, *J* = 8.0 Hz); (2) a trans double bond at *δ*_H_ 7.58 and 6.27 (1H each, d, *J* = 16.0 Hz); (3) a monosubstituted double bond at *δ*_H_ 5.22 (1H, dd, *J* = 10.8, 1.2 Hz), 5.26 (1H, dd, *J* = 17.6, 1.2 Hz) and 5.93 (1H, dd, *J* = 17.6, 10.8 Hz); (4) an olefinic proton at *δ*_H_ 5.10 (1H, tt, *J* = 7.2, 1.6 Hz); (5) two anomeric protons at *δ*_H_ 4.43 (1H, d, *J* = 8.0 Hz) and 5.18 (1H, d, *J* = 1.6 Hz); (6) two methylene groups at *δ*_H_ 2.04 (2H, m), 1.58, 1.62 (1H each, m), and four methyl groups at *δ*_H_ 1.67, 1.60, 1.39 (3H each, s), and 1.08 (3H, d, *J* = 6.4 Hz). The ^13^C NMR spectrum of **1** (Table 2) showed a carbonyl at *δ*_C_ 168.3, three double bonds at *δ*_C_ 114.7–148.0, a benzene ring at *δ*_C_ 115.2–149.8, two anomeric carbons at *δ*_C_ 99.4 and 103.1, nine sugar carbons at *δ*_C_ 62.5–82.0, a quaternary carbon at *δ*_C_ 81.6, two methylene groups at *δ*_C_ 23.6 and 42.6, and four methyl groups at *δ*_C_ 17.7–25.9. The above ^1^H and ^13^C NMR features of **1** were related closely to those of linaloyl-(3-*O*-*α*-l-rhamnopyranosyl)-(4-*O*-*trans*-*p*-coumaroyl)-*β*-d-glucopyranoside (lipedoside B-III) [21], except that the 4-substituted phenyl in lipedoside B-III was replaced by the 3,4-disubstituted phenyl in **1**. The acid hydrolysis experiment of **1** gave d-glucose and l-rhamnose identified by TLC. Furthermore, the HMBC experiment of **1** (Figure 2) displayed the long-distance correlations: between *δ*_H_ 4.33 (H-1′ of glucosyl) and *δ*_C_ 81.6 (C-3 of aglycone), between *δ*_H_ 5.18 (H-1″ of rhamnosyl) and *δ*_C_ 82.0 (C-3′ of glucosyl), between *δ*_H_ 7.58 (H-7′′′ of caffeoyl) and *δ*_C_ 127.6 (C-1′′′ of caffeoyl), and between *δ*_H_ 4.89 (H-4′ of glucosyl) and *δ*_C_ 168.3 (carbonyl of caffeoyl). In addition, the enzymatic hydrolysis experiment of **1** gave (*R*)-linalool and (*S*)-linalool (4:6). The ^1^H and ^13^C NMR signals of **1** were assigned by ^1^H-^1^H COSY, HSQC and HMBC experiments (Appendix A). Based on the above evidence, compound **1** was characterized as a mixture (*R*:*S* = 4:6) of 3(*R*)- and 3(*S*)-linaloyl-(3-*O*-*α*-l-rhamnopyranosyl)-(4-*O*-*trans*-caffeoyl)-*O*-*β*-d-glucopyranoside. It is a novel monoterpenoid glycoside, named ligurobustoside T.

Compound **2** was determined as C_31_H_44_O_12_ by HRESIMS (*m*/*z* 631.2728 [M + Na]^+^, calculated 631.2730 for C_31_H_44_NaO_12_). The ^1^H and ^13^C NMR data of **2** (Table 1 and Table 2) were similar to those of **1**, except the 4-*O*-*trans*-caffeoyl in **1** was replaced by a *trans*-*p*-coumaroyl [*δ*_H_ 6.81, 7.45 (2H each, d, *J* = 8.8 Hz)] at a different position in **2**. The acid hydrolysis experiment of **2** gave d-glucose and l-rhamnose identified by TLC. The HMBC experiment of **2** (Figure 2) showed the long-distance correlations: between *δ*_H_ 4.39 (H-1′ of glucosyl) and *δ*_C_ 81.5 (C-3 of aglycone), between *δ*_H_ 5.17 (H-1″ of rhamnosyl) and *δ*_C_ 84.4 (C-3′ of glucosyl), and between *δ*_H_ 4.30, 4.45 (H-6′ of glucosyl) and *δ*_C_ 169.0 (carbonyl of coumaroyl). Additionally, the enzymatic hydrolysis experiment of **2** gave (*R*)-linalool and (*S*)-linalool (4:6). The ^1^H and ^13^C NMR signals of **2** were assigned by ^1^H-^1^H COSY, HSQC and HMBC experiments (Appendix A). Thus, compound **2** was confirmed as a mixture (*R*:*S* = 4:6) of 3(*R*)- and 3(*S*)-linaloyl-(3-*O*-*α*-l-rhamnopyranosyl)-(6-*O*-*trans*-*p*-coumaroyl)-*O*-*β*-d-glucopyranoside, which is a new monoterpenoid glycoside and named ligurobustoside T_1_.

Compound **3** was analyzed as C_31_H_44_O_13_ by HRESIMS (*m*/*z* 647.2680 [M + Na]^+^, calculated 647.2680 for C_31_H_44_NaO_13_). The ^1^H and ^13^C NMR data of **3** (Table 1 and Table 2) are similar to those of **2** except for some data of the aglycone. The HSQC experiment of **3** displayed the correlations between *δ*_H_ 4.78 (H-8a of aglycone), 4.88 (H-8b of aglycone) and *δ*_C_ 111.4 (C-8 of aglycone), meaning that the C-6 double bond in **2** was replaced by the C-7 double bond in **3**. The ^1^H-^1^H COSY experiment of **3** (Figure 2) displayed the correlations between *δ*_H_ 1.22 (H-4 of aglycone), 3.95 (H-6 of aglycone) and *δ*_H_ 1.60 (H-5 of aglycone), meaning that a hydroxyl was linked at C-6 in **3**. Thus, the aglycone of **3** was 3,7-dimethyl-octa-1,7-diene-3,6-diol. The acid hydrolysis experiment of **3** gave d-glucose and l-rhamnose identified by TLC. The HMBC experiment of **3** (Figure 2) displayed the long-distance correlations: between *δ*_H_ 4.38 (H-1′ of glucosyl) and *δ*_C_ 81.4 (C-3 of aglycone), between *δ*_H_ 5.17 (H-1″ of rhamnosyl) and *δ*_C_ 84.4 (C-3′ of glucosyl), and between *δ*_H_ 4.30, 4.45 (H-6′ of glucosyl) and *δ*_C_ 169.0 (carbonyl of coumaroyl). The ^1^H and ^13^C NMR signals of **3** were assigned by ^1^H-^1^H COSY, HSQC and HMBC experiments (Appendix A). Therefore, compound **3** was determined to be 3-(3,6-dihydroxy-3,7-dimethyl-octa-1,7-dienyl)-(3-*O*-*α*-l-rhamnopyranosyl)-(6-*O*-*trans*-*p*-coumaroyl)-*O*-*β*-d-glucopyranoside. It is a novel monoterpenoid glycoside named ligurobustoside T_2_.

Compound **4** was analyzed as C_3__1_H_46_O_1__4_ by HRESIMS (*m*/*z* 665.2784 [M + Na]^+^, calculated 665.2785 for C_31_H_46_NaO_14_). The NMR spectra of **4** showed two stereoisomers **4a** and **4b** (2:1). The ^1^H NMR spectrum of **4a** (Table 1) displayed the following signals: (1) a 4-substituted phenyl at *δ*_H_ 6.81, 7.46 (2H each, d, *J* = 8.8 Hz); (2) a trans double bond at *δ*_H_ 6.34, 7.64 (1H each, d, *J* = 16.0 Hz); (3) a monosubstituted double bond at *δ*_H_ 5.19 (1H, dd, *J* = 10.8, 2.0 Hz), 5.24 (1H, dd, *J* = 18.0, 2.0 Hz) and 5.92 (1H, dd, *J* =18.0, 10.8 Hz); (4) two anomeric protons at *δ*_H_ 4.41 (1H, d, *J* = 8.0 Hz), 5.18 (1H, d, *J* = 2.0 Hz); (5) a methenyl at *δ*_H_ 3.21 (1H, dd, *J* = 10.4, 2.0 Hz); (6) two methylene groups at *δ*_H_ 1.32–1.90 (4H, m); (7) four methyl groups at *δ*_H_ 1.11, 1.14, 1.36 (3H each, s), 1.25 (3H, d, *J* = 6.4 Hz). The ^13^C NMR spectrum of **4a** (Table 2) revealed a carbonyl at *δ*_C_ 169.0, two double bonds at *δ*_C_ 115.0–146.8, a 4-substituted phenyl at *δ*_C_ 116.9–161.4, two anomeric carbons at *δ*_C_ 99.4 and 102.7, nine sugar carbons at *δ*_C_ 64.9–84.2, two quaternary carbons at *δ*_C_ 73.9 and 81.5, a methenyl at *δ*_C_ 80.1, two methylene groups at *δ*_C_ 26.4 and 39.9, and four methyl groups at *δ*_C_ 17.9–25.8. The above ^1^H and ^13^C NMR data of **4a** were similar to those of 3-(6,7-dihydroxy-3,7-dimethyloct-1-enyl)-(3-*O*-*α*-l-rhamnopyranosyl)-(4-*O*-*trans*-*p*-coumaroyl)-*O*-*β*-d-glucopyranoside (lipedoside B-VI) [21], except the *trans*-*p*-coumaroyl was linked at different positions. The acid hydrolysis experiment of **4** gave d-glucose and l-rhamnose identified by TLC. The HMBC experiment of **4a** (Figure 2) displayed the long-distance correlations: between *δ*_H_ 4.41 (H-1′ of glucosyl) and *δ*_C_ 81.5 (C-3 of aglycone), between *δ*_H_ 5.18 (H-1″ of rhamnosyl) and *δ*_C_ 84.2 (C-3′ of glucosyl), and between *δ*_H_ 4.30, 4.45 (H-6′ of glucosyl) and *δ*_C_ 169.0 (carbonyl of coumaroyl). The ^1^H and ^13^C NMR signals of **4** were assigned by ^1^H-^1^H COSY, HSQC and HMBC experiments (Appendix A). So, **4a** was identified as 3-(3,6,7-trihydroxy-3,7-dimethyloct-1-enyl)-(3-*O*-*α*-l-rhamnopyranosyl)-(6-*O*-*trans*-*p*-coumaroyl)-*O*-*β*-d-glucopyranoside.

The NMR data of **4b** (Table 1 and Table 2) are similar to those of **4a**, except the *trans*-*p*-coumaroyl in **4a** was replaced by the *cis*-*p*-coumaroyl (*δ*_H_ 6.87, 5.78 (1H each, d, *J* = 12.8 Hz, H-7′′′, H-8′′′)) in **4b**. The HMBC experiment of **4b** (Figure 2) showed the long-distance correlations: between *δ*_H_ 4.36 (H-1′ of glucosyl) and *δ*_C_ 81.5 (C-3 of aglycone), between *δ*_H_ 5.15 (H-1″ of rhamnosyl) and *δ*_C_ 84.2 (C-3′ of glucosyl), and between *δ*_H_ 4.25, 4.40 (H-6′ of glucosyl) and *δ*_C_ 168.1 (carbonyl of coumaroyl). So, **4b** was identified as 3-(3,6,7-trihydroxy-3,7-dimethyloct-1-enyl)-(3-*O*-*α*-l-rhamnopyranosyl)-(6-*O*-*cis*-*p*-cou-maroyl)-*O*-*β*-d-glucopyranoside. In conclusion, compound **4** is a mixture of novel monoterpenoid glycosides **4a** and **4b**, named ligurobustoside T_3-__4_.

Compound **5** was analyzed as C_32_H_46_O_13_ by HRESIMS (*m*/*z* 661.2833 [M + Na]^+^, calculated 661.2836 for C_32_H_46_NaO_13_). The ^1^H and ^13^C NMR data of **5** (Table 1 and Table 2) are similar to those of **2** except for some data of the aglycone. The ^1^H-^1^H COSY experiment of **5** (Figure 2) displayed the correlations between *δ*_H_ 2.36 (2H, d, *J* = 7.2 Hz, H-4 of aglycone), 5.40 (1H, d, *J* = 16.0 Hz, H-6 of aglycone) and *δ*_H_ 5.64 (1H, dt, *J* = 16.0, 7.2 Hz, H-5 of aglycone), meaning that the C-6 double bond in **2** was replaced by the C-5(*E*) double bond in **5**. The HMBC experiment of **5** (Figure 2) displayed the correlation between *δ*_H_ 3.09 (OCH_3_) and *δ*_C_ 76.5 (C-7 of aglycone). Hence, the aglycone of **5** was (5*E*)-7-methoxy-3,7-dimethyl-octa-1,5-dien-3-ol. The acid hydrolysis experiment of **5** gave d-glucose and l-rhamnose identified by TLC. The HMBC experiment of **5** (Figure 2) displayed the long-distance correlations: between *δ*_H_ 4.41 (H-1′ of glucosyl) and *δ*_C_ 81.2 (C-3 of aglycone), between *δ*_H_ 5.17 (H-1″ of rhamnosyl) and *δ*_C_ 84.2 (C-3′ of glucosyl), and between *δ*_H_ 4.32, 4.45 (H-6′ of glucosyl) and *δ*_C_ 168.9 (carbonyl of coumaroyl). The ^1^H and ^13^C NMR signals of **5** were assigned by ^1^H-^1^H COSY, HSQC and HMBC experiments (Appendix A). Therefore, compound **5** was determined to be (5*E*)-3-(3-hydroxy-7-methoxy-3,7-dimethyl-octa-1,5-dienyl)-(3-*O*-*α*-l-rhamnopyranosyl)-(6-*O*-*trans*-*p*-coumaroyl)-*O*-*β*-d-glucopyranoside. It is a novel monoterpenoid glycoside, named ligurobustoside T_5_.

Compound **6** was determined as C_37_H_54_O_16_ by HRESIMS (*m*/*z* 777.3312 [M + Na]^+^, calculated 777.3310 for C_37_H_54_NaO_16_). The ^1^H and ^13^C NMR data of **6** (Table 1 and Table 2) are similar to those of lipedoside B-III [21], except there was another rhamnosyl in **6**. The acid hydrolysis experiment of **6** yielded d-glucose and l-rhamnose identified by TLC. The HMBC experiment of **6** (Figure 2) showed the long-distance correlations: between *δ*_H_ 4.44 (H-1′ of glucosyl) and *δ*_C_ 81.6 (C-3 of aglycone), between *δ*_H_ 5.19 (H-1″ of inner rhamnosyl) and *δ*_C_ 81.9 (C-3′ of glucosyl), between *δ*_H_ 5.04 (H-1′′′ of outer rhamnosyl) and *δ*_C_ 81.7 (C-4″ of inner rhamnosyl), and between *δ*_H_ 4.91 (H-4′ of glucosyl) and *δ*_C_ 168.2 (carbonyl of coumaroyl). The ^1^H and ^13^C NMR signals of **6** were assigned by ^1^H-^1^H COSY, HSQC and HMBC experiments (Appendix A). Thus, compound **6** was confirmed as linaloyl-[3-*O*-*α*-l-rhamnopyranosyl-(1→4)-*α*-l-rhamnopyranosyl]-(4-*O*-*trans*-*p*-coumaroyl)-*O*-*β*-d-glucopyranoside, which is a new monoterpenoid glycoside and named ligurobustoside T_6_.

Compound **7** was determined as C_37_H_54_O_16_ by HRESIMS (*m*/*z* 777.3312 [M + Na]^+^, calculated 777.3310 for C_37_H_54_NaO_16_). The ^1^H and ^13^C NMR data of **7** (Table 1 and Table 2) are related closely to those of **6**, except the *trans*-*p*-coumaroyl (*δ*_H_ 7.66, 6.33 (1H each, d, *J* = 16.0 Hz, H-7′′′′, H-8′′′′)) in **6** was replaced by the *cis*-*p*-coumaroyl (*δ*_H_ 6.98, 5.76 (1H each, d, *J* = 12.8 Hz, H-7′′′′, H-8′′′′)) in **7**. The acid hydrolysis experiment of **7** yielded d-glucose and l-rhamnose identified by TLC. The HMBC experiment of **7** (Figure 2) showed the long-distance correlations: between *δ*_H_ 4.41 (H-1′ of glucosyl) and *δ*_C_ 81.6 (C-3 of aglycone), between *δ*_H_ 5.29 (H-1″ of inner rhamnosyl) and *δ*_C_ 79.8 (C-3′ of glucosyl), between *δ*_H_ 5.13 (H-1′′′ of outer rhamnosyl) and *δ*_C_ 80.6 (C-4″ of inner rhamnosyl), and between *δ*_H_ 4.86 (H-4′ of glucosyl) and *δ*_C_ 166.9 (carbonyl of coumaroyl). The ^1^H and ^13^C NMR signals of **7** were assigned by ^1^H-^1^H COSY, HSQC and HMBC experiments (Appendix A). Thus, compound **7** was identified as linaloyl-[3-*O*-*α*-l-rhamnopyranosyl-(1→4)-*α*-l-rhamnopyranosyl]-(4-*O*-*cis*-*p*-coumaroyl)-*O*-*β*-d-glucopyranoside. It is a new monoterpenoid glycoside, named ligurobustoside T_7_.

Compound **8** was analyzed as C_32_H_46_O_13_ by HRESIMS (*m*/*z* 661.2831 [M + Na]^+^, calculated 661.2836 for C_32_H_46_NaO_13_). The NMR spectra of **8** exhibited two stereoisomers **8a** and **8b** (2:1). The ^1^H NMR spectrum of **8a** (Table 1) displayed the following signals: (1) a 4-substituted phenyl at *δ*_H_ 6.80, 7.45 (2H each, d, *J* = 8.4 Hz); (2) two trans double bonds at *δ*_H_ 6.35, 7.64 (1H each, d, *J* = 16.0 Hz), 5.44 (1H, d, *J* = 15.6 Hz), 5.55 (1H, m); (3) an olefinic proton at *δ*_H_ 5.41 (1H, t, *J* = 8.0 Hz); (4) two anomeric protons at *δ*_H_ 4.31 (1H, d, *J* = 8.0 Hz), 5.17 (1H, d, *J* = 2.0 Hz); (5) two methylene groups at *δ*_H_ 4.27 (2H, d, *J* = 8.0 Hz), 2.76 (2H, d, *J* = 10.2 Hz); (6) four methyl groups at *δ*_H_ 1.23, 1.23, 1.65 (3H each, s), 1.24 (3H, d, *J* = 6.4 Hz); and (7) a methoxy at *δ*_H_ 3.12 (3H, s). The ^13^C NMR spectrum of **8a** (Table 2) revealed a carbonyl at *δ*_C_ 169.1, three double bonds at *δ*_C_ 114.8–146.9, a 4-substituted phenyl at *δ*_C_ 117.0–161.7, two anomeric carbons at *δ*_C_ 102.6 and 102.7, nine sugar carbons at *δ*_C_ 64.7–84.0, a quaternary carbon at *δ*_C_ 76.4, two methylene groups at *δ*_C_ 66.3, 43.5, a methoxy at *δ*_C_ 50.6, and four methyl groups at *δ*_C_ 16.6-26.2. The above ^1^H and ^13^C NMR data of **8a** were similar to those of (2*E*,5*E*)-1-(1,7-dihydroxy-3,7-dimethyl-2,5-octa- dienyl)-(3-*O*-*α*-l-rhamnopyranosyl)-(4-*O*-*trans*-*p*-coumaroyl)-*O*-*β*-d-glucopyranoside (ligurobustoside I) [8], except the *trans*-*p*-coumaroyl was linked at different positions, and there was another methyl in **8a**. The HMBC experiment of **8a** (Figure 2) showed the correlation between *δ*_H_ 3.12 (OCH_3_) and *δ*_C_ 76.4 (C-7 of aglycone). The NOEDS experiment of **8a** (Figure 2) displayed the correlation between *δ*_H_ 5.41 (H-2 of aglycone) and *δ*_H_ 2.76 (H-4 of aglycone). Therefore, the aglycone of **8a** was (2*E*,5*E*)-7-methoxy- 3,7-dimethyl-octa-2,5-dien-1-ol. The acid hydrolysis experiment of **8** gave d-glucose and l-rhamnose identified by TLC. The HMBC experiment of **8a** (Figure 2) displayed the long-distance correlations: between *δ*_H_ 4.31 (H-1′ of glucosyl) and *δ*_C_ 66.3 (C-1 of aglycone), between *δ*_H_ 5.17 (H-1″ of rhamnosyl) and *δ*_C_ 84.0 (C-3′ of glucosyl), and between *δ*_H_ 4.35, 4.50 (H-6′ of glucosyl) and *δ*_C_ 169.1 (carbonyl of coumaroyl). The ^1^H and ^13^C NMR signals of **8** were assigned by ^1^H-^1^H COSY, HSQC and HMBC experiments (Appendix A). Consequently, the structure of **8a** was determined to be (2*E*,5*E*)-1-(1-hydroxy-7-methoxy-3,7-dimethyl-octa-2,5-dienyl)-(3-*O*-*α*-l-rhamnopyranosyl)-(6-*O*- *trans*-*p*-coumaroyl)-*O*-*β*-d-glucopyranoside.

The NMR data of **8b** (Table 1 and Table 2) are similar to those of **8a**, except the *trans*-*p*-coumaroyl in **8a** was replaced by the *cis*-*p*-coumaroyl (*δ*_H_ 6.87, 5.79 (1H each, d, *J* = 12.8 Hz, H-7′′′, H-8′′′)) in **8b**. The HMBC experiment of **8b** (Figure 2) displayed the long-distance correlations: between *δ*_H_ 4.27 (H-1′ of glucosyl) and *δ*_C_ 66.3 (C-1 of aglycone), between *δ*_H_ 5.16 (H-1″ of rhamnosyl) and *δ*_C_ 84.0 (C-3′ of glucosyl), and between *δ*_H_ 4.31, 4.48 (H-6′ of glucosyl) and *δ*_C_ 168.1 (carbonyl of coumaroyl). So, **8b** was identified as (2*E*,5*E*)-1-(1-hydroxy-7-methoxy-3,7-dimethyl-octa-2,5-dienyl)-(3-*O*-*α*-l-rhamnopyranosyl)-(6-*O*-*cis*-*p*-coumaroyl)-*O*-*β*-d-glucopyranoside. In conclusion, compound **8** is a mixture of novel monoterpenoid glycosides **8a** and **8b**, named ligurobustoside T_8–9_.

Compounds **9**–**11** (NMR data see Appendix A) were identified as ligurobustosides G (**9a**) and H (**9b**), ligurobustoside C (**10**), ligurobustosides K (**11a**) and L (**11b**), respectively, by direct comparison with published spectral data (^1^H, ^13^C NMR) [8,9].

### 3.2. The Bioactivities of Compounds ***1**–**11***

Compounds **1**–**11** from the leaves of *L. robustum* were tested for the inhibitory activities on FAS, *α*-glucosidase, *α*-amylase, and the antioxidant effects. The results of bioactivity assays are shown in Table 3. As shown in Table 3, compound **2** revealed stronger FAS inhibitory activity (IC_50_: 2.36 ± 0.10 μM) than the positive control orlistat (IC_50_: 4.46 ± 0.13 μM); compound **2** showed weaker *α*-glucosidase inhibitory effect than the positive control acarbose; compounds **2**–**6**, **8**, **9** and **11** displayed weaker *α*-amylase inhibitory effect than the positive control acarbose; compounds **1**, **2**, **5** and **11** exhibited more potent ABTS radical scavenging activity (IC_50_: 6.91 ± 0.10~9.41 ± 0.22 μM) than the positive control L-(+)-ascorbic acid (IC_50_: 10.06 ± 0.19 μM), while compound **1** displayed weaker DPPH radical scavenging activity (IC_50_: 19.74 ± 0.23 μM) than L-(+)-ascorbic acid (IC_50_: 13.66 ± 0.13 μM).

Because FAS is a potential therapeutic target for anti-obesity drugs [17,18,19], compounds **2**, **6** and **10** with strong FAS inhibitory activity might be a part of the constituents with anti-obesity activity in *L. robustum*. In addition, the results suggested that the FAS inhibitory activity would reduce or disappear when the monoterpene unit of glycoside was substituted with hydroxyl, or the *trans*-*p*-coumaroyl of glycoside was replaced by other groups.

## 4. Conclusions

In summary, the phytochemical research on the leaves of *L. robustum* resulted in the separation of eleven monoterpenoid glycosides (**1**–**11**), including eight new compounds (**1**–**8**) identified with spectroscopic method (^1^H, ^13^C NMR, ^1^H-^1^H COSY, HSQC, HMBC, NOEDS, HRESIMS), and physical and chemical methods. The biological study showed that compound **2** revealed stronger FAS inhibitory activity (IC_50_: 2.36 ± 0.10 μM) than the positive control orlistat (IC_50_: 4.46 ± 0.13 μM); compounds **1**, **2**, **5** and **11** displayed more potent ABTS radical scavenging activity (IC_50_: 6.91 ± 0.10~9.41 ± 0.22 μM) than the positive control L-(+)-ascorbic acid (IC_50_: 10.06 ± 0.19 μM); compound **2** revealed also moderate *α*-glucosidase and *α*-amylase inhibitory activities. This study provided a theoretical basis for the leaves of *L. robustum* as a functional tea to treat obesity.

## Figures and Tables

**Figure 1 molecules-27-03709-f001:**
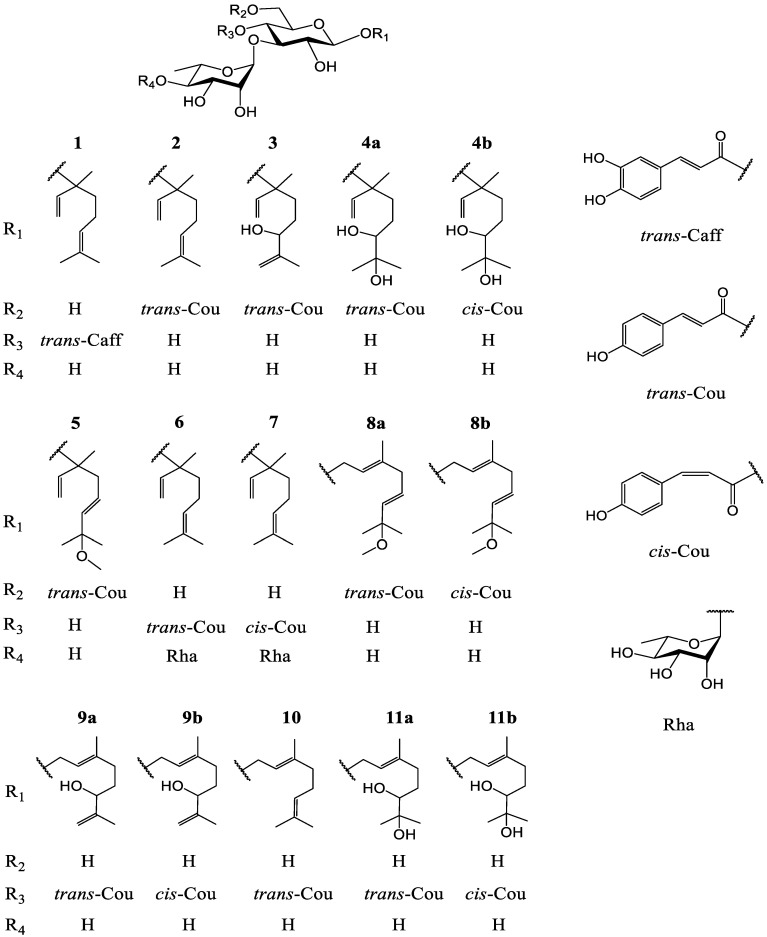
Structures of compounds **1**–**11** from the leaves of *L. robustum*.

**Figure 2 molecules-27-03709-f002:**
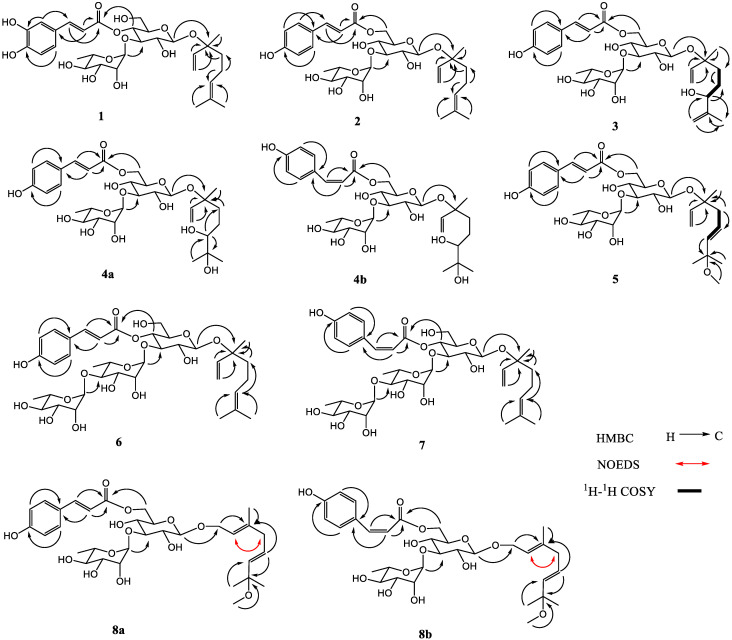
Key HMBC, ^1^H-^1^H COSY and NOEDS correlations of compounds **1**–**8**.

**Table 1 molecules-27-03709-t001:** ^1^H NMR data of compounds **1**–**8** from *L. robustum* in CD_3_OD *^a^*.

**No**	**1 *^b^***	**2 *^b^***	**3 *^c^***	**4a *^b^***	**4b *^b^***
1	5.22 dd (10.8, 1.2)	5.19 dd (10.8, 1.2)	5.19 br. d (10.8)	5.19 dd (10.8, 2.0)	5.19 dd (10.8, 2.0)
	5.26 dd (17.6, 1.2)	5.23 dd (18.0, 1.2)	5.24 br. d (18.0)	5.24 dd (18.0, 2.0)	5.24 dd (18.0, 2.0)
2	5.93 dd (17.6, 10.8)	5.90 dd (18.0, 10.8)	5.90 dd (18.0, 10.8)	5.92 dd (18.0, 10.8)	5.92 dd (18.0, 10.8)
4	1.58 m	1.56 m	1.22 dd (10.8, 2.4)	1.57 m	1.57 m
	1.62 m	1.60 m		1.90 m	1.90 m
5	2.04 m	2.02 m	1.60 m	1.32 m	1.32 m
				1.70 m	1.70 m
6	5.10 tt (7.2, 1.6)	5.07 tt (7.2, 1.2)	3.95 m	3.21 dd (10.4, 2.0)	3.21 dd (10.4, 2.0)
8	1.67 s	1.62 br. s	4.78 br. s	1.11 s	1.11 s
			4.88 br. s		
9	1.60 s	1.56 br. s	1.67 s	1.14 s	1.14 s
10	1.39 s	1.34 s	1.35 s	1.36 s	1.36 s
7-OCH_3_					
Glc					
1′	4.43 d (8.0)	4.39 d (8.0)	4.38 d (8.4)	4.41 d (8.0)	4.36 d (8.0)
2′	3.36 m	3.29 m	3.29 m	3.29 m	3.27 m
3′	3.77 t (9.2)	3.49 m	3.48 m	3.50 t (8.8)	3.46 t (8.8)
4′	4.89 m	3.34 m	3.32 m	3.33 m	3.29 m
5′	3.45 m	3.49 m	3.48 m	3.47 m	3.42 m
6′	3.49 m	4.30 dd (12.0, 6.8)	4.30 dd (12.0, 6.6)	4.30 dd (12.0, 6.0)	4.25 dd (12.0, 6.0)
	3.57 m	4.45 dd (12.0, 2.4)	4.45 dd (12.0, 2.4)	4.45 dd (12.0, 2.4)	4.40 dd (12.0, 2.4)
inner- Rha					
1″	5.18 d (1.6)	5.17 d (2.0)	5.17 d (1.8)	5.18 d (2.0)	5.15 d (2.0)
2″	3.91 dd (3.2, 1.6)	3.94 dd (3.2, 2.0)	3.94 m	3.94 dd (3.2, 2.0)	3.94 dd (3.2, 2.0)
3″	3.58 m	3.70 dd (9.6, 3.2)	3.70 dd (9.6, 3.6)	3.70 dd (9.6, 3.2)	3.70 dd (9.6, 3.2)
4″	3.29 t (9.6)	3.40 t (9.6)	3.39 t (9.6)	3.39 t (9.6)	3.39 t (9.6)
5″	3.56 m	4.00 dd (9.6, 6.4)	4.00 m	3.99 dd (9.6, 6.4)	3.99 dd (9.6, 6.4)
6″	1.08 d (6.4)	1.25 d (6.4)	1.24 d (6.6)	1.25 d (6.4)	1.24 d (6.4)
outer- Rha					
1′′′					
2′′′					
3′′′					
4′′′					
5′′′					
6′′′					
Ester					
2′′′′	7.05 d (2.0)	7.45 d (8.8)	7.46 d (8.4)	7.46 d (8.8)	7.64 d (8.8)
3′′′′		6.81 d (8.8)	6.80 d (8.4)	6.81 d (8.8)	6.76 d (8.8)
5′′′′	6.77 d (8.0)	6.81 d (8.8)	6.80 d (8.4)	6.81 d (8.8)	6.76 d (8.8)
6′′′′	6.95 dd (8.0, 2.0)	7.45 d (8.8)	7.46 d (8.4)	7.46 d (8.8)	7.64 d (8.8)
7′′′′	7.58 d (16.0)	7.64 d (16.0)	7.64 d (16.2)	7.64 d (16.0)	6.87 d (12.8)
8′′′′	6.27 d (16.0)	6.33 d (16.0)	6.33 d (16.2)	6.34 d (16.0)	5.78 d (12.8)
**No**	**5 *^b^***	**6 *^b^***	**7 *^b^***	**8a *^b^***	**8b *^b^***
1	5.19 dd (10.8, 1.2)	5.23 dd (10.8, 1.6)	5.22 dd (10.8, 1.6)	4.27 d (8.0)	4.27 d (8.0)
	5.22 dd (17.6, 1.2)	5.25 dd (17.6, 1.6)	5.24 dd (17.6, 1.6)		
2	5.90 dd (17.6, 10.8)	5.93 dd (17.6, 10.8)	5.92 dd (17.6, 10.8)	5.41 t (8.0)	5.41 t (8.0)
4	2.36 d (7.2)	1.58 m	1.58 m	2.76 d (10.2)	2.76 d (10.2)
		1.62 m	1.62 m		
5	5.64 dt (16.0, 7.2)	2.05 m	2.04 m	5.55 m	5.55 m
6	5.40 d (16.0)	5.10 m	5.10 m	5.44 d (15.6)	5.44 d (15.6)
8	1.20 s	1.67 s	1.67 s	1.23 s	1.23 s

9	1.20 s	1.60 s	1.60 s	1.23 s	1.23 s
10	1.33 s	1.39 s	1.38 s	1.65 s	1.65 s
7-OCH_3_	3.09 s			3.12 s	3.12 s
Glc					
1′	4.41 d (8.0)	4.44 d (7.6)	4.41 d (8.0)	4.31 d (8.0)	4.27 d (8.0)
2′	3.31 m	3.37 m	3.37 m	3.30 m	3.28 m
3′	3.50 t (8.8)	3.77 t (9.6)	3.77 t (9.6)	3.51 m	3.46 m
4′	3.35 m	4.91 t (9.6)	4.86 t (9.6)	3.37 m	3.33 m
5′	3.49 m	3.46 m	3.46 m	3.51 m	3.47 m
6′	4.32 dd (12.0, 7.2)	3.50 m	3.50 m	4.35 dd (12.0, 6.0)	4.31 dd (12.0, 6.0)
	4.45 dd (12.0, 2.0)	3.57 m	3.57 m	4.50 dd (12.0, 2.0)	4.48 dd (12.0, 2.0)
inner- Rha					
1″	5.17 d (2.0)	5.19 d (2.0)	5.29 d (2.0)	5.17 d (2.0)	5.16 d (2.0)
2″	3.94 dd (3.6, 2.0)	3.86 dd (3.2, 2.0)	3.82 dd (3.2, 2.0)	3.94 m	3.92 m
3″	3.70 dd (9.6, 3.6)	3.68 dd (9.6, 3.2)	3.68 dd (9.6, 3.2)	3.70 dd (9.6, 3.2)	3.68 dd (9.6, 3.2)
4″	3.40 t (9.6)	3.39 m	3.45 m	3.40 m	3.40 m
5″	4.00 dd (9.6, 6.4)	3.59 m	3.60 m	4.00 dd (9.6, 6.4)	4.00 dd (9.6, 6.4)
6″	1.25 d (6.4)	1.08 d (6.0)	1.21 d (6.4)	1.24 d (6.4)	1.23 d (6.4)
outer- Rha					
1′′′		5.04 d (2.0)	5.13 d (2.0)		
2′′′		3.90 dd (3.2, 2.0)	3.82 dd (3.2, 2.0)		
3′′′		3.51 m	3.51 m		
4′′′		3.32 m	3.34 m		
5′′′		3.46 m	3.46 m		
6′′′		1.04 d (6.0)	1.21 d (6.4)		
Ester					
2′′′′	7.45 d (8.4)	7.48 d (8.4)	7.72 d (8.4)	7.45 d (8.4)	7.65 d (8.4)
3′′′′	6.81 d (8.4)	6.82 d (8.4)	6.77 d (8.4)	6.80 d (8.4)	6.75 d (8.4)
5′′′′	6.81 d (8.4)	6.82 d (8.4)	6.77 d (8.4)	6.80 d (8.4)	6.75 d (8.4)
6′′′′	7.45 d (8.4)	7.48 d (8.4)	7.72 d (8.4)	7.45 d (8.4)	7.65 d (8.4)
7′′′′	7.63 d (16.0)	7.66 d (16.0)	6.98 d (12.8)	7.64 d (16.0)	6.87 d (12.8)
8′′′′	6.32 d (16.0)	6.33 d (16.0)	5.76 d (12.8)	6.35 d (16.0)	5.79 d (12.8)

*^a^* Coupling constants (*J* values in Hz) are shown in parentheses. *^b^* At 400 MHz. *^c^* At 600 MHz.

**Table 2 molecules-27-03709-t002:** ^13^C NMR data of compounds **1**–**8** from *L. robustum* in CD_3_OD.

**No**	**1 *^a^***	**2 *^a^***	**3 *^a^***	**4a *^a^***	**4b *^a^***
1	115.9	115.7	115.8	115.9	115.9
2	144.3	144.3	144.3	144.3	144.3
3	81.6	81.5	81.4	81.5	81.5
4	42.6	42.5	30.2	39.9	39.9
5	23.6	23.6	30.1	26.4	26.4
6	125.7	125.7	76.9	80.1	80.1
7	132.2	132.1	148.7	73.9	73.9
8	25.9	25.8	111.4	24.9	24.9
9	17.7	17.7	17.7	25.8	25.8
10	23.2	23.5	23.5	23.9	23.9
7-OCH_3_					
Glc					
1′	99.4	99.3	99.4	99.4	99.3
2′	76.3	75.7	75.8	75.8	75.8
3′	82.0	84.4	84.4	84.2	84.2
4′	70.7	70.8	70.8	70.7	70.7
5′	75.7	75.0	75.1	75.1	75.0
6′	62.5	65.0	64.9	64.9	62.7
inner-Rha					
1″	103.1	102.8	102.8	102.7	102.4
2″	72.4	72.4	72.4	72.4	72.4
3″	72.0	72.3	72.3	72.3	72.3
4″	73.8	74.0	74.0	74.0	74.0
5″	70.4	70.0	70.0	70.0	70.0
6″	18.5	17.9	17.9	17.9	17.9
outer-Rha					
1′′′					
2′′′					
3′′′					
4′′′					
5′′′					
6′′′					
Ester					
1′′′′	127.6	127.1	126.9	127.1	127.6
2′′′′	115.2	131.1	131.2	131.2	133.8
3′′′′	146.8	116.9	117.0	116.9	115.9
4′′′′	149.8	161.5	161.9	161.4	160.2
5′′′′	116.5	116.9	117.0	116.9	115.9
6′′′′	123.2	131.1	131.2	131.2	133.8
7′′′′	148.0	146.7	148.7	146.8	145.3
8′′′′	114.7	115.0	114.8	115.0	116.2
CO	168.3	169.0	169.0	169.0	168.1
**No**	**5 *^a^***	**6 *^b^***	**7 *^b^***	**8a *^a^***	**8b *^a^***
1	116.0	115.9	115.9	66.3	66.3
2	144.0	144.3	144.3	122.3	122.3
3	81.2	81.6	81.6	140.9	140.9
4	45.5	42.6	42.6	43.5	43.5
5	127.4	23.6	23.7	129.3	129.3
6	139.2	125.7	125.7	138.1	138.1
7	76.5	132.2	132.2	76.4	76.4
8	26.1	25.9	25.9	26.2	26.2
9	26.1	17.7	17.7	26.2	26.2
10	23.5	23.2	23.1	16.6	16.6
7-OCH_3_	50.7			50.6	50.6
Glc					
1′	99.3	99.4	99.4	102.6	102.6
2′	75.8	76.3	76.5	75.6	75.6
3′	84.2	81.9	79.8	84.0	84.0
4′	70.8	70.6	70.4	70.5	70.4
5′	75.0	75.7	75.6	75.5	75.4
6′	64.9	62.4	62.5	64.7	64.5
inner-Rha					
1″	102.8	102.7	101.9	102.7	102.8
2″	72.4	72.7	72.9	72.4	72.4
3″	72.3	72.9	73.0	72.2	72.2
4″	74.0	81.7	80.6	74.0	74.0
5″	70.0	68.9	68.6	70.0	70.3
6″	17.9	19.2	18.9	17.9	17.9
outer-Rha					
1′′′		103.5	103.2		
2′′′		72.3	72.3		
3′′′		72.3	72.3		
4′′′		73.8	73.9		
5′′′		70.3	70.3		
6′′′		17.7	17.8		
Ester					
1′′′′	127.1	127.0	127.5	126.9	127.5
2′′′′	131.2	131.4	134.3	131.2	133.8
3′′′′	116.9	117.0	116.0	117.0	116.0
4′′′′	161.4	161.5	160.3	161.7	160.3
5′′′′	116.9	117.0	116.0	117.0	116.0
6′′′′	131.2	131.4	134.3	131.2	133.8
7′′′′	146.7	147.5	147.4	146.9	145.3
8′′′′	115.0	114.8	115.8	114.8	116.2
CO	168.9	168.2	166.9	169.1	168.1

*^a^* At 100 MHz. *^b^* At 150 MHz.

**Table 3 molecules-27-03709-t003:** The results of bioactivity assays of compounds **1**–**11** from *L. robustum ^a^*.

Compounds	FAS IC_50_ (μM) *^b^*	*α*-Glucosidase Inhibition at 0.1 mM (%)	*α*-Amylase Inhibition at 0.1 mM (%)	DPPH IC_50_ (μM) *^b^*	ABTS^•+^ IC_50_ (μM) *^b^*
**1**	NA *^c^*	NA	NA	19.74 ± 0.23 b	6.91 ± 0.10 a
**2**	2.36 ± 0.10 a	48.1 ± 4.3 b	31.5 ± 0.5 b	>250	9.41 ± 0.22 c
**3**	21.77 ± 0.38 c	27.3 ± 0.3 c	32.5 ± 6.3 b	NA	16.00 ± 0.69 g
**4**	>100	NA	28.2 ± 3.9 b	NA	9.66 ± 0.17 cd
**5**	23.71 ± 0.45 d	13.8 ± 2.0 d	35.6 ± 2.0 b	NA	6.93 ± 0.01 a
**6**	4.78 ± 0.14 b	12.0 ± 1.7 d	26.1 ± 3.0 b	NA	11.30 ± 0.16 e
**7**	>100	NA	NA	NA	20.21 ± 0.33 j
**8**	25.83 ± 0.47 e	24.7 ± 3.5 c	31.4 ± 1.9 b	NA	19.50 ± 0.46 i
**9**	21.67 ± 0.46 c	12.4 ± 5.6 d	29.2 ± 8.4 b	NA	18.66 ± 0.47 h
**10**	4.68 ± 0.16 b	28.7 ± 2.1 c	NA	NA	15.10 ± 0.10 f
**11**	61.74 ± 0.45 f	NA	31.3 ± 1.3 b	NA	7.92 ± 0.23 b
Orlistat *^d^*	4.46 ± 0.13 b				
Acarbose *^d^*		93.2 ± 0.1 a	51.8 ± 2.5 a		
L-(+)-ascorbic acid *^d^*				13.66 ± 0.13 a	10.06 ± 0.19 d

*^a^* Data are expressed as mean ± SD (*n* = 3). Means with the same letter are not significantly different (one-way analysis of variance, *α* = 0.05). *^b^* IC_50_: the final concentration of sample needed to inhibit 50% of enzyme activity or scavenge 50% of free radical. *^c^* NA: no activity. *^d^* Positive control.

## Data Availability

Not applicable.

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
