# Peer review of "Monoterpenoid Glycosides from the Leaves of *Ligustrum robustum* and Their Bioactivities"

_molecules, 2022, doi:10.3390/molecules27123709_

Round 1
Reviewer 1 Report
I am glad to have an opportunity for reviewing this manuscript. This manuscript deals with monoterpenoid glycosides from Ligustrum robustum leaves. The authors present some novel compounds from the plant and examined their bioactivities to elucidate the bioactive compounds of the plant; therefore, I think that this manuscript may be welcomed by readers. I hope that my opinions could help improve the quality of your study, and my comments for your manuscript are as follows;
- In the Abstract, there are grammatical mistakes in “a functional tea to 15 clear heat, remove toxins, and treat obesity, diabetes, in Southwest China.”
- For readers, please write the experimental methods in detail for the “2.3. Determination of bioactivities.”
- I cannot understand 2.2.2. and 2.2.3. Why did you need to hydrolyze compounds? Did you identify the chemical structures of compounds after hydrolysis? Then, you need to include 2.2.2. and 2.2.3. into 2.2.1. Or did you use hydrolyzed aglycons of each compound when examining their bioactivities? (I think it is not) I strongly recommend supplementing the information in the related parts.
- Please check Table 3 again. You omitted the full name of FAS and “alpha” before -glycosidase and -amylase.
Author Response
- In the Abstract, there are grammatical mistakes in “a functional tea to 15 clear heat, remove toxins, and treat obesity, diabetes, in Southwest China.”
Authors accept the advice and revise as follows: a functional tea to clear heat, remove toxins, and treat obesity and diabetes, in Southwest China.
- For readers, please write the experimental methods in detail for the “2.3. Determination of bioactivities.”
Authors accept the advice and offer the experimental methods in detail (S1) while delete the last sentence of 2.3. (revision version 2.6.).
- I cannot understand 2.2.2. and 2.2.3. Why did you need to hydrolyze compounds? Did you identify the chemical structures of compounds after hydrolysis? Then, you need to include 2.2.2. and 2.2.3. into 2.2.1. Or did you use hydrolyzed aglycons of each compound when examining their bioactivities? (I think it is not) I strongly recommend supplementing the information in the related parts.
In order to identify the structures of monosaccharide residues in glycosides 1-8, as customary, acid hydrolysis of glycosides 1-8 was carried out, and the acquired monosaccharides were identified by TLC with authentic samples (3.1.). Similarly, in order to identify the stereo configuration of aglycon in glycosides 1-2, enzymatic hydrolysis of glycosides 1-2 was carried out, and the stereo configurations of the acquired aglycons were affirmed by specific optical rotation (3.1.). Acid hydrolysis of compounds 1-8 and enzymatic hydrolysis of compounds 1-2 didn’t belong to “extraction and isolation” (2.2.1.), and 2.2.-2.4. are numbered again.
- Please check Table 3 again. You omitted the full name of FAS and “alpha” before -glycosidase and -amylase.
In the introduction (1.), it was noted that FAS was the abbreviation of fatty acid synthase. The font of “a ”(alpha) was symbol.
Reviewer 2 Report
The work described in the present manuscript is consistent with the scope of the journal.
Authors described the isolation and characterization of active compounds from leaves of L. robustum. Their fatty acid synthase, alfa-glucosidase, alfa-amylase, and antioxidant effects have been investigated. Overall results help to understand the anti-obesity properties of the tea.
The work was well conduced and therefor only minor revisions are required prior to a possible publication, specifically:
Major comments:
- I believe that compounds 6 and 7 are stereoisomers and therefore, considering the same criteria used for compounds 4,8, 9 and 11, why not ascribe compounds 6 and 7 as 6a and 6b? for the same reason, the information described in lines 138-147 is the same for both compounds because they are isomers. Please revise these paragraphs accordingly.
- Line 104-106: it would be interesting to determine the yield of the extraction for each compound and the relative abundance of each of the compounds isolated. Please consider the inclusion of this information in the manuscript.
- Section 3.1 (lines 183-356): please always mention the units of the chemical shift together with the values discussed along this section.
- Lines 354-356: Please refer to the respective results section or Table number reported in the ESI. Also, uniformize the assignment of the NMR data for compounds 9-11. It is not logical to describe the assignment of the signals for compound 10 in the form of text and for compounds 9 and 11 in the form of a Table.
- Lines 362-364: the enzymes’ inhibitory effects mentioned are not weaker than that found for the controls. The IC50 is lower, which means that the compounds are more potent than the controls, and therefore stronger (not weaker). Please correct these statements.
- Conclusion: In my opinion, it is important to conclude that compound 2 is the unique with the strongest anti-fatty acid synthase (FAS), alfa-glucosidase, alfa-amylase, and the antioxidant effect, concomitantly This important finding must be highlighted in this section.
Minor comments:
- Please delete some empty spaces in lines16, 192, 380 and 388.
- Lines 39-41: please rewrite this sentence. May be a link between the sentence before is more appropriate.
- Line 75, please mention the ration of MeOH:H2O used.
- Table 1: The coupling constants must be in italic.
- Table 3: replace “DPPH” by “DPPH assay” and “ABTS” by “ABTS assay”; the “alfa” symbol is not present.
- Conclusions / funding: please check the size of the font.
Author Response
Major comments:
- I believe that compounds 6 and 7 are stereoisomers and therefore, considering the same criteria used for compounds 4,8, 9 and 11, why not ascribe compounds 6 and 7 as 6a and 6b? for the same reason, the information described in lines 138-147 is the same for both compounds because they are isomers. Please revise these paragraphs accordingly.
The compounds 1-11 were numbered and discussed by the isolated products. Althought the NMR spectra of compound 4 showed 2 stereoisomers 4a and 4b, compound 4 (15.3 mg) was one isolated product (original version 2.2.1., revision version 2.3.). But compounds 6 (17.6 mg) and 7 (8.2 mg) were two isolated products (original version 2.2.1., revision version 2.3.). If compounds 6 and 7 were numbered as 6a and 6b, and discussed together, meaning that compounds 6 and 7 were one isolated product, which was in contradiction with 2.2.1.(revision version 2.3.).
- Line 104-106: it would be interesting to determine the yield of the extraction for each compound and the relative abundance of each of the compounds isolated. Please consider the inclusion of this information in the manuscript.
Authors accept the advice and supplement the experimental information.
- Section 3.1 (lines 183-356): please always mention the units of the chemical shift together with the values discussed along this section.
In “general experimental procedure” (2.1.) , it was noted that “chemical shifts are expressed in d (ppm) with tetramethylsilane (TMS) as the internal standard”. And it was usual to discuss the structure of compound with the chemical shift values (omitting the unit), for example, Hwang’s article (Molecules 2021, 26, 3164) and Jakimiuk’s article (Molecules 2021, 26, 5631).
- Lines 354-356: Please refer to the respective results section or Table number reported in the ESI. Also, uniformize the assignment of the NMR data for compounds 9-11. It is not logical to describe the assignment of the signals for compound 10 in the form of text and for compounds 9 and 11 in the form of a Table.
Authors accept the advice and supplement the relative figure numbers reported in supporting information. The NMR data of compound 9-11 are listed in Table S1-S3.
- Lines 362-364: the enzymes’ inhibitory effects mentioned are not weaker than that found for the controls. The IC50 is lower, which means that the compounds are more potent than the controls, and therefore stronger (not weaker). Please correct these statements.
In Table 3, FAS inhibitory activity, and DPPH and ABTS radical scavenging activities were reported as IC50, while a-glucosidase inhibitory effect and a-amylase inhibitory effect were reported as inhibition rate (but not IC50). The a-glucosidase inhibition rate of compound 2 was lower than the positive control acarbose.
- Conclusion: In my opinion, it is important to conclude that compound 2 is the unique with the strongest anti-fatty acid synthase (FAS), alfa-glucosidase, alfa-amylase, and the antioxidant effect, concomitantly This important finding must be highlighted in this section.
Authors accept partly the advice and supplement statement: compound 2 revealed also moderate a-glucosidase and a-amylase inhibitory activities.
Minor comments:
- Please delete some empty spaces in lines16, 192, 380 and 388.
Authors accept the advice.
- Lines 39-41: please rewrite this sentence. May be a link between the sentence before is more appropriate.
Authors consider original sentences are better than others, because chemical constitutes and bioactivities expressed respectively are clearer.
- Line 75, please mention the ration of MeOH:H2O used.
It was noted that “eluting with MeOH-H2O at a flow rate of 30 mL/min” at Line 78, and the ratio of MeOH-H2O was noted in 2.2.1. (revision version 2.3.).
- Table 1: The coupling constants must be in italic.
The coupling constants were not in italic, for example, Hwang’s article (Molecules 2021, 26, 3164) and Jakimiuk’s article (Molecules 2021, 26, 5631).
- Table 3: replace “DPPH” by “DPPH assay” and “ABTS” by “ABTS assay”; the “alfa” symbol is not present.
Authors accept partly the advice and the font of “a ”(alpha) is set as symbol.
- Conclusions / funding: please check the size of the font.
Authors accept the advice and the size of the font is set again.
Round 2
Reviewer 1 Report
The authors mended all of the suggestions and comments.